# Urinary CD80 and Serum suPAR as Biomarkers of Glomerular Disease among Adults in Brazil

**DOI:** 10.3390/diagnostics13020203

**Published:** 2023-01-05

**Authors:** Renata de Cássia Zen, Wagner Vasques Dominguez, Ivone Braga, Luciene Machado dos Reis, Lectícia Barbosa Jorge, Luis Yu, Viktoria Woronik, Cristiane Bitencourt Dias

**Affiliations:** 1Nephrology Department, Hospital das Clínicas, Faculty of Medicine, University of São Paulo, São Paulo 01246-903, SP, Brazil; 2Laboratory of Renal Pathophysiology, Hospital das Clínicas, Faculty of Medicine, University of São Paulo, São Paulo 01246-903, SP, Brazil

**Keywords:** urinary CD80, supAR, biomarkers, minimal change disease, focal segmental glomerulosclerosis

## Abstract

Introduction: Urinary CD80 has been shown to have good specificity for minimal change disease (MCD) in children. However, the investigation of circulating factors such as soluble urokinase plasminogen activator receptor (suPAR) as biomarkers of focal segmental glomerulosclerosis (FSGS) is quite controversial. The objective of this study was to determine whether urinary CD80 and serum suPAR can be used for the diagnosis of MCD and FSGS, respectively, in the adult population of Brazil. We also attempted to determine whether those biomarkers assess the response to immunosuppressive treatment. Methods: This was a prospective study in which urine and blood samples were collected for analysis of CD80 and suPAR, respectively, only in the moment of renal biopsy, from patients undergoing to diagnostic renal biopsy. At and six months after biopsy, we analyzed serum creatinine, serum albumin, and proteinuria in order to evaluate the use of the CD80 and suPAR collected in diagnosis as markers of response to immunosuppressive treatment. In healthy controls were collected urinary CD80 and proteinuria, serum suPAR, and creatinine. Results: The results of 70 renal biopsies were grouped, by diagnosis, as follows: FSGS (*n* = 18); membranous nephropathy (*n* = 14); MCD (*n* = 5); and other glomerulopathies (*n* = 33). There was no significant difference among the groups in terms of the urinary CD80 levels, and serum suPAR was not significantly higher in the FSGS group, as would have been expected. Urinary CD80 correlated positively with nephrotic syndrome, regardless of the type of glomerular disease. Neither biomarker correlated with proteinuria at six months after biopsy. Conclusion: In adults, urinary CD80 can serve as a marker of nephrotic syndrome but is not specific for MCD, whereas serum suPAR does not appear to be useful as a diagnostic or treatment response marker.

## 1. Introduction

The pathogenesis of minimal change disease (MCD) has yet to be fully elucidated, and it remains unclear whether its pathogenesis differs from that of focal segmental glomerulosclerosis (FSGS). In 1974, Shalhoub [1] proposed that the nephrotic syndrome caused by MCD is related to T-cell dysfunction. Decades later, Cara-Fuentes et al. [2] suggested that the dysregulation of T cells in podocyte injury in MCD involves the axis of regulation between CD80 and cytotoxic T-lymphocyte antigen-4 (CTLA-4). Formerly known as B7-1, CD80 is a transmembrane protein present in antigen-presenting cells, natural killer cells, and B cells [3]. It is probably a major regulator of T lymphocytes because it has two modes of action, one stimulating those cells through its effect on the CD28 receptor and the other inhibiting them through its effect on the cytotoxic T-lymphocyte-associated antigen 4 (CTLA-4) receptor [4]. The CTLA-4 surface protein is expressed by T cells and downregulates T-cell activation after binding to CD80 on antigen-presenting cells.

On the basis of the knowledge that the podocyte can, under a given stimulus, acquire characteristics of dendritic (antigen-presenting) cells and express CD80, it has been demonstrated that urinary CD80 levels are higher in pediatric patients with active MCD than in those with MCD that is in remission or those with another glomerular disease [5,6]. In such patients, no increase in serum CD80 has been demonstrated, and the hypothesis that it is a circulating factor has therefore been refuted. The expression of CD80 by the podocyte could also induce the sequestration of essential proteins such as nephrin, CD2-associated protein, and zonula occludens 1 to the podocyte, causing disruption of the slit diaphragm complex, in addition to acting on integrin signaling, which has the well-known function of preserving the integrity of the podocyte–glomerular basement membrane complex [2,7].

In 1972, Hoyer et al. [8] observed cases of immediate recurrence of FSGS after kidney transplantation and proposed that a circulating factor plays a role in the pathophysiology of primary FSGS. Such recurrence has been shown to improve after plasmapheresis, possibly due to the removal of a circulating factor [9,10]. Since then, various studies have been carried out in an attempt to identify a circulating factor in FSGS.

The best-known circulating factor in FSGS is the soluble urokinase plasminogen activator receptor (suPAR). The urokinase plasminogen activator (uPA) system is composed of a protease, a receptor (uPAR), and an inhibitor. The uPAR is a 45–55 kDa protein with three domains (DI, DII, and DIII) linked to glycosylphosphatidylinositol, which binds to the membranes of some immunologically active cells, such as neutrophils, lymphocytes, monocytes, macrophages, activated T cells, endothelial cells, megakaryocytes, tumor cells, and podocytes. In addition, uPAR can bind to various ligands, including uPA, vitronectin, and integrins. That binding leads to cellular activities such as adhesion, migration, differentiation, and proliferation. In podocytes, uPAR is one of the pathways capable of activating the αvβ3 integrin, thus promoting cell motility and the activation of small GTPases, such as Cdc42 and Rac1, which can lead to podocyte contraction, changing the phenotype from stationary to mobile and culminating in the collapse of the podocyte [11].

In animal models, Wei et al. [12] demonstrated that high doses of recombinant suPAR induce alterations in podocyte processes, increased β3 integrin activity, and proteinuria. In humans, Wei et al. [13] also demonstrated that the serum concentration of suPAR was significantly higher in patients with FSGS than in healthy individuals. They observed no significant variation in suPAR among individuals with MCD (in relapse or remission) or among those with membranous nephropathy or preeclampsia. In addition, the authors found that, among individuals with FSGS, pretransplantation serum concentrations of suPAR were higher in those who experienced recurrence after transplantation. Although subsequent studies produced promising results, not all achieved the same results for suPAR, which raised questions regarding its interpretation and even the manner in which it is analyzed in the laboratory.

The aim of this study was to determine whether urinary CD80 and serum suPAR can be used for the diagnosis of MCD and FSGS, respectively, in adults in Brazil. We also attempted to determine whether those biomarkers assess the response to immunosuppressive treatment.

## 2. Methods

This was a prospective study, in which urine and blood samples were collected from patients hospitalized on the Nephrology ward of the Hospital das Clínicas, operated by the University of São Paulo School of Medicine, in the city of São Paulo, Brazil. All of the patients were admitted between January 2018 and January 2020. The study was approved by the Research Ethics Committee of the Hospital das Clínicas (Reference no. 73117917.2.0000.0068), and all participating patients gave written informed consent.

### 2.1. Inclusion Criteria

We included patients aged ≥ 14 years admitted to the nephrology ward with a first clinical presentation suggestive of glomerular disease for diagnostic renal biopsy. 

### 2.2. Exclusion Criteria

Patients receiving immunosuppressive therapy for more than 30 days were excluded, as were those with positive serology for hepatitis B, hepatitis C, or HIV.

### 2.3. Baseline Patient Data

For the patients hospitalized for renal biopsy, the standard departmental protocol was followed, including the analysis of serum and urinary creatinine by the kinetic colorimetric method; serum albumin by the colorimetric method; serum C-reactive protein (normal value < 5 mg/L) by immunoturbidimetry and 24 h proteinuria or urinary protein level determined by the turbidimetric method, the latter for calculation of the protein-to-creatinine ratio. Creatinine clearance was calculated with the Chronic Kidney Disease–Epidemiology Collaboration equation (CKDEPI) [14]. In addition, the patients underwent serology for hepatitis B, hepatitis C, or HIV, as well as an autoimmune panel and immunofixation to aid in the etiological diagnosis. Demographic data were collected at the time of the diagnosis, which was based on the biopsy findings.

Blood and urine samples were centrifuged in a refrigerated centrifuge, separated into aliquots, and stored at −80 °C. Subsequently, urinary CD80 and serum suPAR were analyzed in the aliquots. An enzyme-linked immunosorbent assay (ELISA Kit; Bender MedSystems, Vienna, Austria) [6] was used in order to measure urinary CD80, and the results were adjusted for urinary creatinine as recommended by the literature. A similar assay (Quantikine Human uPAR ELISA kit; R&D Systems, Minneapolis, MN, USA) was used for the assessment of serum suPAR.

Statistical analyzes were performed with patients with MCD and FSGS as separate pathologies, but also together in a subgroup of podocytopathies.

### 2.4. Baseline Control Data

To create a control group, we recruited ten healthy individuals from among graduate students and residents in the glomerulopathy group at the hospital. From those ten individuals, blood and urine samples were collected. The samples were treated and stored in the same way as those collected from the patients, after which they were used for the analysis of serum and urinary creatinine, as well as for the measurement of urinary CD80 and serum suPAR. Proteinuria was evaluated in an isolated sample to calculate the protein-to-creatinine ratio.

### 2.5. Data Assessed at Six Months after Renal Biopsy

At six months after renal biopsy, patient data (serum creatinine, serum albumin, 24 h proteinuria, and protein-to-creatinine ratio) were collected from the electronic medical records. The six-month data were correlated with urinary CD80 and serum suPAR values at the time of renal biopsy in order to evaluate the use of the latter as markers of response to immunosuppressive treatment.

### 2.6. Statistical Analysis

Continuous variables are expressed as mean ± standard deviation for samples with normal distribution or as median and interquartile range (IQR) for those without. Categorical variables are expressed as absolute and relative frequencies. Differences among three or more groups were assessed with one-way analysis of variance, unless the sample was not normally distributed, in which case the Kruskal–Wallis test was used. Between-group comparisons of categorical variables were made with the chi-square test, and linear correlations were evaluated with the Pearson or Spearman test, as appropriate. Values of *p* < 0.05 were considered statistically significant. All statistical tests were performed with GraphPad Prism software, version 9.0 (GraphPad Software Inc., San Diego, CA, USA).

## 3. Results

During the study period, we evaluated 70 patients, as well as the ten individuals in the control group. Among the patients, the median (IQR) age was 39 (14–75) years, the median (IQR) serum creatinine was 1.28 (0.45–8.50) mg/dL, creatinine clearance of 55 (5–146) mL/min/1.73 m^2^, the median (IQR) serum albumin was 2.55 (0.90–4.60) g/dL, and the median (IQR) urinary protein-to-creatinine ratio was 3.46 (0.19–19) g/g. Of the 70 patients, 43 (61.4%) were women. For analysis, the diagnoses, based on the renal biopsy findings, were grouped as follows: FSGS (*n* = 18); membranous nephropathy (*n* = 14); MCD (*n* = 5); and other glomerulopathies (*n* = 33). The other glomerulopathies group comprised cases of immunoglobulin A nephropathy (*n* = 12); lupus nephritis (*n* = 7); membranoproliferative glomerulonephritis (*n* = 1); diabetic nephropathy (*n* = 4); multiple myeloma (*n* = 2); amyloidosis (*n* = 5); hypertensive nephrosclerosis (*n* = 1) and antiphospholipid antibody syndrome (*n* = 1). In the control group, the median (IQR) age was 32.5 (25–49) years, the median (IQR) serum creatinine was 1.11 (0.82–1.27) mg/dL, and the median (IQR) protein/urinary creatinine ratio was 0.003 (0–0.04) g/g. Of the ten controls, seven (70%) were women.

As can be seen in Table 1, the median (IQR) urinary CD80 adjusted for urinary creatinine was 104 (19.70–369.60) ng/g in patients with MCD, 63.15 (30.50–244.60) ng/g in those with FSGS, 76.80 (31.22–402.20) ng/g in those with membranous nephropathy, and 24.70 (15.10–41.40) ng/g in the controls, with no statistical difference between the glomerulopathies except against control groups (*p* = 0.048). The median (IQR) serum suPAR (Table 1) was 3887 (2359–4620) pg/mL in the patients with FSGS, 3266 (2887–4225) pg/mL in those with MCD, 3091 (2018–3711) pg/mL in those with membranous nephropathy, and 1336 (1033–1586) pg/mL in the controls, with no statistically significant difference between the patients with FSGS and those with MCD or membranous nephropathy, although there was a significant difference between the patients with FSGS and the controls (*p* = 0.0001).

When the patients with MCD were grouped together with the patients with FSGS, a podocytopathy group, and compared with the patients with membranous glomerulopathy, other glomerulopathies and control group (Table 2), the median (IQR) urinary CD80 adjusted for urinary creatinine, 80 (33–266.60) ng/g, was significantly higher in comparison with other glomerulopathies and the control group (*p* = 0.005). For serum suPAR when the patients with FSGS were grouped together with those with MCD (Table 2), the median (IQR) serum suPAR was 3597 (2424–4531) pg/mL, which was also statistically different but only in relation to the control group (*p* < 0.0001). 

As illustrated in Figure 1 and Figure 2, urinary CD80 adjusted for urinary creatinine correlated negatively with baseline serum albumin in all patients, regardless of glomerulopathy (*r* = −0.5, *p* < 0.0001), whereas it correlated positively, albeit weakly, with baseline proteinuria (*r* = 0.31, *p* = 0.006). When the patients were divided into two groups by serum albumin level (<3.5 g/dL and ≥3.5 g/dL), the median (IQR) urinary CD80 adjusted for urinary creatinine was 77.27 (33.50–190) ng/g in the <3.5 g/dL group (*n* = 53) and 14.9 (7.83–42.15) ng/g in the ≥3.5 g/dL group (*n* = 17), and the difference was statistically significant (*p* < 0.0001). However, no such difference was found when the patients were divided into two groups by proteinuria (<3.5 g/day and ≥3.5 g/day). 

When the patients were divided into two groups by serum creatinine (<1.2 mg/dL and ≥1.2 mg/dL), the median (IQR) serum suPAR was found to be 3066 (2184–3409) pg/mL in the <1.2 mg/dL group and 4065 (2935–5405) pg/mL in the ≥1.2 mg/dL group, and the difference was statistically significant (*p* = 0.0006). 

Among the patients with FSGS (*n* = 18), urinary CD80 and serum suPAR did not differ significantly between those with collapsing FSGS (*n* = 9) and those with other forms of FSGS (*n* = 9). The median (IQR) urinary CD80 was 91 (35.95–254.10) ng/g in those with collapsing FSGS and 46.30 (18–254) ng/g in those with other forms (*p* = 0.42). The median (IQR) serum suPAR was 3278 (2138–4709) pg/mL in the patients with collapsing FSGS and 3972 (3470–5235) pg/mL in those with other forms (*p* = 0.29).

In the analysis of the 70 patients, we also found a positive correlation between serum suPAR and urinary CD80 with serum C-reactive protein, with *p* = 0.017 and *r* = 0.3 and *p* = 0.04 and *r* = 0.25, respectively.

At 6 months after diagnosis by renal biopsy, the patients in the podocitopathy group’s median serum creatinine was 1.00 (0.66–1.45), the serum albumin was 3.8 (2.80–4.30), and the urinary protein-to-creatinine ratio was 1.57 (0.18–3.47). The urinary CD80 in this group did not correlate with baseline creatinine or with proteinuria and serum creatinine at 6 months. In the same way, serum suPAR did not correlate with proteinuria, at baseline or at 6 months after diagnosis, although it did show a positive correlation with baseline creatinine (*r* = 0.56, *p* = 0.0052; Figure 3) and without correlation with creatinine at 6 months after diagnosis. 

## 4. Discussion

In recent years, research has been carried out with the objective of identifying serum and urinary biomarkers, the determination of which is less invasive, that could replace renal biopsy for the differential diagnosis of glomerulopathies. In addition to facilitating the diagnosis, such biomarkers are also expected to provide information on the type of response to immunosuppressive therapy, as has been demonstrated in membranous nephropathy.

In MCD studies involving populations consisting predominantly of children, Cara-Fuentes et al. [15], Liao et al. [16], and Mishra et al. [17] found urinary CD80 to be a relevant diagnostic biomarker. However, a study involving pediatric and adult patients showed that urinary CD80 is elevated in various glomerulopathies that result in a urinary protein-to-creatinine ratio ≥ 3 g/g [18]. In that study, urinary CD80 was elevated even in patients with lupus nephritis who had high levels of proteinuria. Similarly, we did not identify CD80 values high enough to justify its use as a diagnostic marker of MCD in adults. Nevertheless, we found that urinary CD80 correlated positively with proteinuria and negatively with serum albumin, suggesting that it might be useful as a marker of podocyte damage.

In an immunohistochemistry study of renal tissue from patients with type II diabetes and diabetic nephropathy, Fiorina et al. found CD80 positivity in 47% of these patients against zero in control undergoing nephrectomy for renal cancer [19]. In a study involving patients with Fabry disease, which is not a disease with high proteinuria, the degree of podocyturia and urinary CD80 levels were significantly higher among the patients than among the controls, although there was no correlation between podocyturia and urinary CD80 [20].

In a study conducted by Novelli et al. [21], CD80 was evaluated by immunofluorescence and immunoperoxidase staining of renal tissue obtained by biopsy from patients with MCD or FSGS. The authors found traces of CD80 in the renal tissue of patients in both groups, with no difference between the patients with active disease and those in remission, and they observed no tissue positivity in the control group. Although that finding was not considered significant, CD80 was clearly detectable in the presence of inflammatory cells. In that same study, the authors assessed CD80 in the renal tissue of mice with adriamycin-induced FSGS and found no difference between the mice with FSGS and the control mice. Those authors examined the nonspecificity of CD80 as a disease marker and the possibility that it is present in podocytes, albeit at levels below the detection limit. In the present study, we did not evaluate CD80 in renal tissue, because our objective was to identify biomarkers in blood and urine.

We believe that urinary CD80 shows promise for the diagnosis of MCD in children but not in adults. However, a study involving children in Japan produced results similar to ours, showing no difference in urinary CD80 among MCD, FSGS, and other nephrotic syndromes, although also detecting a positive correlation between urinary CD80 and proteinuria, regardless of the type of glomerulopathy [22].

Primary FSGS is probably related to circulating factors, and several have been cited as biomarkers of this disease, although suPAR is one of the most widely studied and investigated in the literature. After the promising results obtained by Wei C et al. [12,13] and Chebotareva N et al. [23], a variety of studies did not find the determination of serum suPAR to be useful in FSGS [24,25]. Evidence that serum suPAR is elevated in patients with impaired glomerular filtration [25] discourages its use in glomerular disease. The results found for serum suPAR in the present study also do not support its utility as a biomarker. Our finding that serum suPAR showed a significant positive correlation only with serum creatinine, which may indicate a more severe kidney disease, albeit without specificity, is in keeping with those of other studies [24,25].

The study conducted by Harel et al. [26] raises a question about the kit used for the determination of the serum level of suPAR. The authors evaluated kits that assess suPAR as an intact molecule comprising DI, DII, and DIII, in comparison with a kit that assesses only DI. They found that the assessment of suPAR by DI effectively distinguishes FSGS from other glomerulopathies and health, as well as showing higher titers in patients with recurrence after renal transplantation. The kit used in our study evaluates intact suPAR, therefore leaving open the possibility of its use.

In a study conducted in China, urinary CD80 was evaluated in children with MCD or FSGS [27]. The authors stratified the patients by urinary CD80 level (>328 ng/g and ≤328 ng/g) and found that renal survival at approximately 60 months of follow-up was better in the >328 ng/g group. The best explanation for that finding is that most of the children with MCD were in the >328 ng/g group, which would therefore be the group with the best treatment response. In our study, no correlation we found of urinary CD80 with proteinuria or serum creatinine at 6 months after renal biopsy. However, there is a need for studies with longer follow-up periods and larger patient samples in order to draw better conclusions regarding the prognostic data.

Prognostic assessments of serum suPAR have been conducted in critically ill patients with a variety of renal diseases, especially sepsis, based on the idea that this biomarker correlates with the activity of the immune system. Elevated serum suPAR levels have been correlated with mortality in intensive care unit patients, although some studies have suggested that there would be no additional gain in relation to more common inflammatory markers such as C-reactive protein [28]. As in the literature, our study showed a positive correlation between serum suPAR and C-reactive protein, as well as urinary CD80 being also positively correlated with this inflammatory marker.

The fact that we studied adult patients at a public hospital in Brazil allowed us to assess information that is still scarce in the literature. However, that also resulted in a relatively small sample of patients with MCD, which makes it difficult to draw definitive conclusions regarding that disease.

Although we did not find urinary CD80 and serum suPAR to be relevant biomarkers diagnostic or assess the response to immunosuppressive treatment of MCD or FSGS, the fact that we evaluated them in various glomerulopathies, as well as in healthy individuals, allowed us to identify a series of correlations, such as those suggesting that urinary CD80 is a marker of podocyte injury and that suPAR is nonspecific in glomerulopathies. Therefore, our results could facilitate future research by improving understanding of the pathophysiology of these diseases.

## 5. Conclusions

It seems feasible to employ urinary CD80 as a marker of nephrotic syndrome in adults, given that it appears to correlate with serum albumin and proteinuria, albeit lacking specificity for MCD. However, serum suPAR not only has minimal specificity for the diagnosis of FSGS but also appears to have no utility in proteinuric glomerular disease or in their assessing the response to immunosuppressive treatment.

## Figures and Tables

**Figure 1 diagnostics-13-00203-f001:**
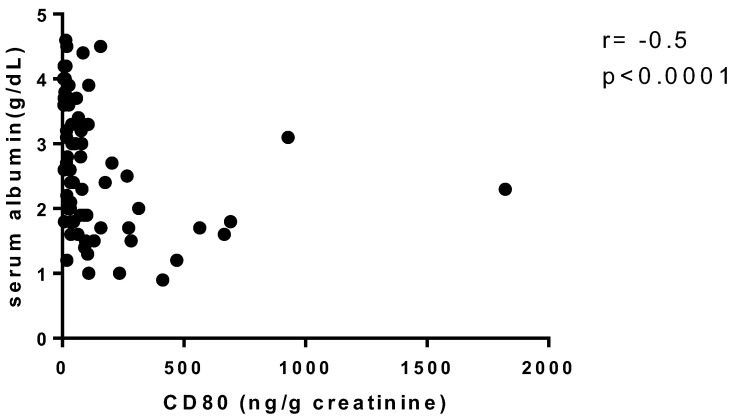
Correlation between urinary CD80 adjusted for urinary creatinine and baseline serum albumin.

**Figure 2 diagnostics-13-00203-f002:**
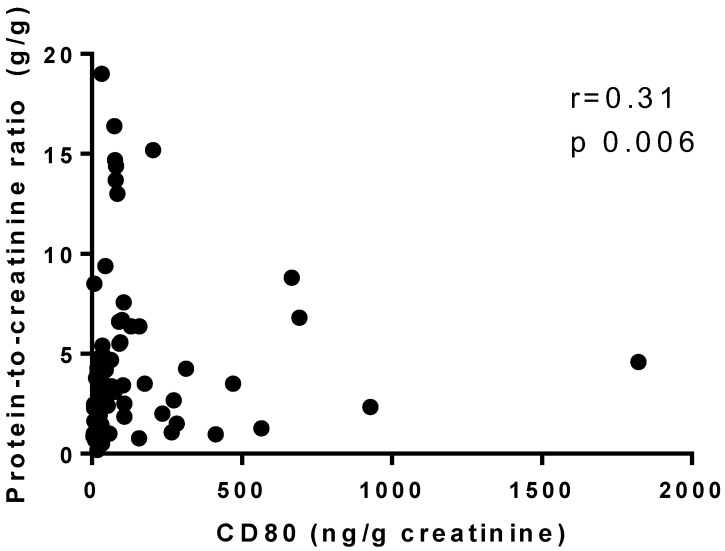
Correlation between urinary CD80 adjusted for urinary creatinine and baseline proteinuria.

**Figure 3 diagnostics-13-00203-f003:**
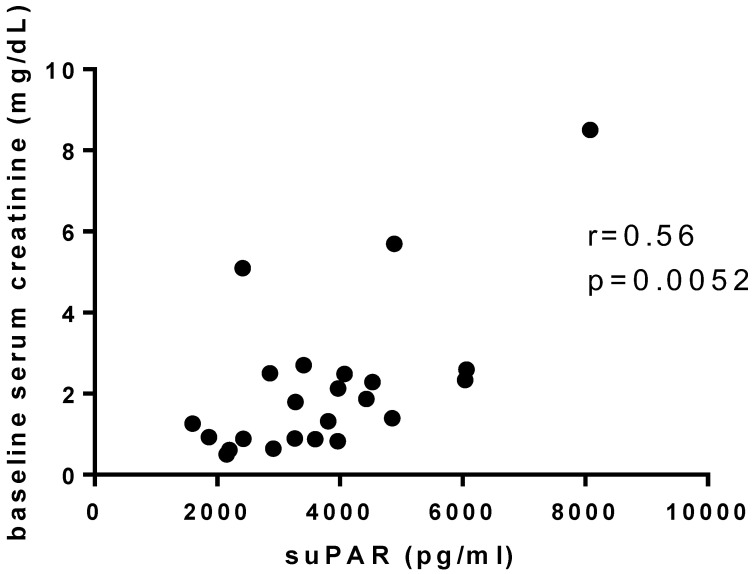
Correlation between serum soluble urokinase plasminogen activator receptor (suPAR) and baseline serum creatinine in Podocitopathy group (suPAR, soluble urokin).

**Table 1 diagnostics-13-00203-t001:** Demographic characteristics and biomarker data at diagnosis in patients with glomerulopathies and in healthy controls.

Characteristic	MCD	FSGS	Membranous Nephropathy	Control	*p* Value
	(*n* = 5)	(*n* = 18)	(*n* = 14)	(*n* = 10)	
Age (years), mean ± SD	37.60 ± 15	33.67 ± 13.60	42.15 ± 19.64	33.60 ± 6.63	0.63
Male sex, *n* (%)	1 (20.0)	10 (55.6)	6 (42.9)	3 (30.0)	<0.0001
Serum creatinine (mg/dL), median (IQR)	0.90 (0.76–1.95)	2 (0.92–2.62)	0.65 (0.50–1.17)	1.11 (0.98–1.18)	0.086
Protein-to-creatinine ratio (g/g), median (IQR)	3.10 (1.04–3.46)	4.05 (1.87–5.44)	4.42 (2.38–7.14)	0.003 (0–0.22)	<0.0001
Serum albumin (g/dL), mean ± SD	2.08 ± 0.99	2.06 ± 0.87	2.25 ± 0.54	-	0.10
CD80 (ng/g creatinine), median (IQR)	104 (19.70–369.60)	63.15 (30.50–244.60)	76.80 (31.22–402.20)	24.70 (15.10–41.40)	0.048 ^‡^
suPAR (pg/mL), median (IQR)	3266 (2887–4225)	3887 (2359–4620)	3091 (2018–3711)	1336 (1033–1586)	0.0001 *

FSGS, focal segmental glomerulosclerosis; IQR, interquartile range; MCD, minimal change disease; SD, standard deviation; suPAR, soluble urokinase plasminogen activator receptor. ^‡^ MCD vs control; * FSGS vs. control.—Serum albumin was not measure in the control group.

**Table 2 diagnostics-13-00203-t002:** Demographic characteristics and biomarker data at diagnosis in the combined Minimal Change Disease + Focal Segmental Glomerulosclerosis group, in comparison with the membranous nephropathy, other glomerulopathies and control groups.

Characteristic	MCD + FSGS	Membrane Nephropathy	Other Glomerulopathies	Control	*p* Value
	(*n* = 23)	(*n* = 14)	(*n* = 33)	(*n* = 10)	
Age (years), mean ± SD	34.52 ± 13.65	42.15 ± 19.64	41.04 ± 15.68	33.60 ± 6.63	0.24
Male sex, *n* (%)	11 (47.8)	6 (42.9)	8 (28.6)	3 (30.0)	0.49
Serum creatinine (mg/dL), median (IQR)	1.80 (0.89–2.50)	0.65 (0.50–1.17)	1.39 (0.72–2.10)	1.11 (0.98–1.18)	0.16
Protein-to-creatinine ratio (g/g), median (IQR)	3.50 (1.50–4.90)	4.42 (2.38–7.14)	2.11 (1.01–5.32)	0.003 (0–0.22)	<0.0001
CD80 (ng/g creatinine), median (IQR)	80 (33–266.6)	76.80 (31.22–402.20)	27.15 (14.53–76.55)	24.70 (15.10–41.40)	0.005 *
suPAR (pg/mL), median (IQR)	3597 (2424–4531)	3091 (2018–3711)	3148 (2431–4015)	1336 (1033–1586)	<0.0001 ^†^

FSGS, focal segmental glomerulosclerosis; IQR, interquartile range; MCD, minimal change disease; SD, standard deviation; suPAR, soluble urokinase plasminogen activator receptor. * MCD + FSGS vs. other glomerulopathies and control. ^†^ MCD + FSGS vs. control.

## Data Availability

The study did not report any data.

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
