# Peer review of "Urinary CD80 and Serum suPAR as Biomarkers of Glomerular Disease among Adults in Brazil"

_diagnostics, 2023, doi:10.3390/diagnostics13020203_

Round 1

Reviewer 1 Report

The paper presented for review concerns the use of urinary CD80 and serum suPAR as biomarkers of glomerular disease. The search for new biomarkers for so-called civilization diseases is extremely important. The presented work is very interesting. Before accepting it for publication, the reviewer would like to ask a few questions.

1. Did the patients aged 14-18 decide to participate in the study themselves, or did you obtain the consent of your legal guardians? You indicate adults in the title. There is a slight inconsistency here. Please explain.

2. Please add an explanation why patients with immunosuppressive therapy for more than 30 days were excluded, as were those with positive serology for hepatitis B, hepatitis C, or HIV.

3. Please add a reference in the place of "An enzyme-linked immunosorbent assay (ELISA Kit; Bender MedSystems, Vienna, Austria) was used in order to measure urinary CD80, and the results were adjusted for urinary creatinine as recommended by the literature"

4. How were healthy volunteers determined to be really healthy? Was a medical interview conducted? What criteria were taken into account?

5. Serum albumin values ​​for healthy volunteers are not shown in Table 1. Was the value undetermined or was it equal to 0?

Author Response

Dear reviewer,

Thank you for your comments and I will forward the answers to your questions.

I will be at your disposal for any other questions.

Best regards,

Renata Zen

Reviewer 2 Report

The manuscript is interesting and well-written.

Author Response

Dear reviewer,

Thank you for your review.

Best regards,

Renata Zen

Reviewer 3 Report

The introduction builds a logical case and context for the problem statement and the problem statement is clear and well articulated. The literature review is up-to-date and the number of references is appropriate and their selection is judicious. The study addresses important problems or issues; the study is worth doing. The research design is defined and clearly described, and is sufficiently detailed to permit the study to replicated. The population is defined clearly and is sufficiently detailed to permit the study to be replicated.  The subjects are appropriate to the research question.

Data analysis procedures are sufficiently described, and are sufficiently detailed to permit the study to be replicated. The conclusions are clearly stated; key points stand out.

The results of this study differ from that of reference 5 & 6 which demonstrated that urinary CD80 levels are higher in pediatric patients with active MCD than in those with MCD that is in remission or those with another glomerular disease. The small sample size     [FSGS (n = 18); membranous nephropathy (n = 14); MCD (n = 5); 153 and other glomerulopathies (n = 33)] id regarded as small. The study limitation regarding the small sample size is not discussed and may have contributed to the overall findings.

Thus the authors need to discuss the limitation(s) of the study.

Author Response

Dear reviewer,

Thank you for your comments and I forward the answer to your suggestion.

I will be at your disposal for any other questions.

Best regards,

Renata Zen
